# Impact of Nitisinone on the Cerebrospinal Fluid Metabolome of a Murine Model of Alkaptonuria

**DOI:** 10.3390/metabo12060477

**Published:** 2022-05-25

**Authors:** Andrew S. Davison, Brendan P. Norman, Hazel Sutherland, Anna M. Milan, James A. Gallagher, Jonathan C. Jarvis, Lakshminarayan R. Ranganath

**Affiliations:** 1Department of Clinical Biochemistry and Metabolic Medicine, Liverpool Clinical Laboratories, Liverpool University Hospitals NHS Foundation Trust, Liverpool L69 3GA, UK; anna.milan@liverpoolft.nhs.uk (A.M.M.); lrang@liverpool.ac.uk (L.R.R.); 2Department of Musculoskeletal and Ageing Science, Institute of Life Course and Medical Sciences, University of Liverpool, Liverpool L7 8TX, UK; bnorman@liverpool.ac.uk (B.P.N.); h.sutherland@liverpool.ac.uk (H.S.); j.a.gallagher@liverpool.ac.uk (J.A.G.); 3School of Exercise Science, Liverpool John Moores University, Liverpool L3 3AF, UK; j.c.jarvis@ljmu.ac.uk

**Keywords:** Alkaptonuria, hypertyrosinaemia, neurotransmitters, cognitive function

## Abstract

Background: Nitisinone-induced hypertyrosinaemia is well documented in Alkaptonuria (AKU), and there is uncertainty over whether it may contribute to a decline in cognitive function and/or mood by altering neurotransmitter metabolism. The aim of this work was to evaluate the impact of nitisinone on the cerebrospinal fluid (CSF) metabolome in a murine model of AKU, with a view to providing additional insight into metabolic changes that occur following treatment with nitisinone. Methods: 17 CSF samples were collected from BALB/c *Hgd*^−/−^ mice (*n* = 8, treated with nitisinone—4 mg/L and *n* = 9, no treatment). Samples were diluted 1:1 with deionised water and analysed using a 1290 Infinity II liquid chromatography system coupled to a 6550 quadrupole time-of-flight mass spectrometry (Agilent, Cheadle, UK). Raw data were processed using a targeted feature extraction algorithm and an established in-house accurate mass retention time database. Matched entities (±10 ppm theoretical accurate mass and ±0.3 min retention time window) were filtered based on their frequency and variability. Experimental groups were compared using a moderated t-test with Benjamini–Hochberg false-discovery rate adjustment. Results: L-Tyrosine, N-acetyl-L-tyrosine, γ-glutamyl-L-tyrosine, p-hydroxyphenylacetic acid, and 3-(4-hydroxyphenyl)lactic acid were shown to increase in abundance (log2 fold change 2.6–6.9, 3/5 were significant *p* < 0.05) in the mice that received nitisinone. Several other metabolites of interest were matched, but no significant differences were observed, including the aromatic amino acids phenylalanine and tryptophan, and monoamine metabolites adrenaline, 3-methoxy-4-hydroxyphenylglycol, and octopamine. Conclusions: Evaluation of the CSF metabolome of a murine model of AKU revealed a significant increase in the abundance of a limited number of metabolites following treatment with nitisinone. Further work is required to understand the significance of these findings and the mechanisms by which the altered metabolite abundances occur.

## 1. Introduction

The widespread metabolic complication of nitisinone therapy in Alkaptonuria (AKU) [1,2,3,4,5,6,7,8] and Hereditary Tyrosinaemia type 1 (HT1) [9,10,11,12] is an increased concentration of circulating L-tyrosine, known as hypertyrosinaemia.

Much of the literature available on hypertyrosinaemia relates to the clinical consequences that have been observed, for example corneal keratopathy and skin rashes [3,4,13,14,15]. These particular side effects typically resolve when nitisinone treatment is stopped or the dosage modified. A more contentious side effect of hypertyrosinaemia has been linked to alterations in neurotransmitter metabolism, specifically in HT1, where it is estimated that ~35% children have altered cognitive function following nitisinone therapy [16]. The reason for this interest is that both dopamine (DP) and serotonin play a pivotal role in cognitive function and mood.

Several mechanisms relating hypertyrosinaemia to altered neurotransmitter metabolism have been proposed, including: (i) increased transport of L-tyrosine into the brain, leading to increased DP in the central nervous system (CNS); (ii) decreased transport of other large neutral aromatic amino acids via the large neutral amino acid transporter 1 (LAT-1) into the brain (i.e., tryptophan), leading to decreased serotonin in the CNS, and (iii) altered serotonin metabolism due to the direct inhibition of tryptophan hydroxylase (TPH; EC 1.14.16.4) activity (rate-limiting step in serotonin biosynthesis) by L-tyrosine [17,18,19].

Thimm et al. [17], in a study investigating neurotransmitter metabolites, showed low concentrations of 5-hydroxyindole acetic acid (5-HIAA, serotonin metabolite) in the cerebrospinal fluid (CSF) of three patients with HT1; however, no direct measurements of DP or serotonin were made in the brain tissue or CSF. Hillgartner et al. [18] later showed in a murine model of HT1 that urinary homovanillic acid (HVA, DP metabolite) increased four-fold in mice receiving treatment with nitisinone, but concluded that this was likely to be a reflection of peripheral breakdown of catecholamines and did not reflect changes in the CNS. More recently, Barone et al. [19] demonstrated that human L-tyrosine hydroxylase isoform 1 and human tryptophan hydroxylase 2, expressed in *Escherichia coli (BL21)*, exhibited prominent substrate inhibition kinetics, and that enzyme activity decreased at elevated L-tyrosine levels. This supports the earlier supposition, but again in this study no downstream metabolites were measured (i.e., DP or serotonin).

There is no direct evidence of altered cognition or central neurotransmitter metabolism in AKU patients with hypertyrosinaemia. In order to assess central neurotransmitter metabolism, either CSF or a brain biopsy are required, as they directly reflect neurotransmitter metabolism in the CNS. This is an accepted practice in the assessment of neurotransmitter disorders [20], but to obtain these samples would be impractical in patients with AKU, owing to their complex musculoskeletal comorbidities, and unethical in the absence of clinical evidence of altered cognition or mood. Previous studies of patients with AKU have made assessments of neurotransmitter metabolism in urine and reported increased urinary excretion of the DP metabolite 3-methoxytyramine (3-MT) in patients receiving nitisinone on a short- (four weeks, 1–8 mg) and long-term (2 mg daily, two years) basis [21,22,23]. Urinary normetadrenaline (NMA) was also shown to significantly decrease in the short-term study at all doses (1–8 mg) [21]. Both studies concluded that these changes are likely to reflect altered peripheral catecholamine metabolism. This is supported by recent findings that no cognitive decline or increased severity of depression were observed in adults on long-term, low-dose nitisinone therapy [24].

A study in a murine model of AKU has also reassuringly demonstrated that monoamine metabolite patterns in brain tissue did not change following treatment with nitisinone, with the exception of the trace amine tyramine, which increased 25-fold [25]. This was the first direct evidence showing that DP and serotonin do not change following treatment with nitisinone. Herein, for the first time, we report the direct analysis of CSF from the same murine model of AKU, using liquid chromatography quadrupole time of flight mass spectrometry (LC-QTOF-MS) analysis and an established in-house accurate mass retention time (AMRT) database [23,26,27]. The rationale for analysing CSF is that it bathes the tissues of the CNS, and may give additional insight into changes in neurotransmitters and their respective metabolites and confirm previous findings [25]. Moreover, it will allow for changes in the wider CSF metabolome to be assessed following treatment with nitisinone.

## 2. Results

### Quality Control and Murine CSF Data

In the absence of biological group QCs, only system QC samples were run. Table 1 demonstrates that the analytical run was acceptable, as the raw peak abundance for all metabolites was <12% CV, and the mean retention time (RT) and ppm error were <0.1 min and <1 ppm, respectively. In addition, reference ion mass calibration and binary pump pressures were very reproducible (Appendix A).

A total of 174 and 135 metabolites were aligned across all CSF samples in positive and negative polarities (for matched compounds, see Metabolomics Workbench: https://www.metabolomicsworkbench.org/, Study ID: adavison_20220505_020022_mwtab.txt and data track ID: 3248, accessed 5 May 2022), respectively, using an in-house AMRT database. After filtering entities based on their frequency (i.e., present in 100% of samples in either group group) and variability across the experimental groups (CV < 25%), 82 and 51 entities were retained from positive and negative polarity profiling experiments, respectively. Of these, five were shown to have a log2 fold change (FC) > 2.0, 3 of these entities were shown to be significantly different (*p* < 0.05), respectively (Table 2).

Clear separation was observed between the CSF metabolite profiles from mice in the two treatment groups (Figure 1, component 1) in positive polarity. Table 2 shows compounds that were significantly different (*p* < 0.05 with a log2 FC > 2.0) between the BALB/c *Hgd*^−/−^ mice that did and did not receive nitisinone therapy.

The abundance of γ-L-glutamyl-L-tyrosine and N-acetyl-L-tyrosine were greater in mice that received nitisinone; log2 FC was 6.9 and 5.0, respectively (Figure 2). However, no *p* value was calculated for these two metabolites, as the CV for raw signal abundance was >25% in both experimental groups.

Several other metabolites of interest (Figure 3) were matched in positive and negative polarities, respectively, with a raw abundance CV < 25% in at least 1 experimental group. These metabolites had a log2 FC < 2.0, and no significant differences were observed between the experimental groups. Of particular relevance in this study were the large neutral aromatic amino acids (LNAAs) tryptophan and phenylalanine, and the catecholamine metabolites octopamine (trace amine metabolite from tyramine) and 3-methoxy-4-hydroxyphenylglycol (MHPG, major metabolite of noradrenaline), which were not significantly different in the two experimental groups.

Other metabolites of interest related to catecholamine metabolism included 5-HIAA and dihydroxyphenylacetic acid (DOPAC). These, however, had raw signal abundance CVs of 53 and 64%, 100 and 61% in the mice that were and were not treated with nitisinone, respectively. These metabolites were therefore excluded from further statistical analysis. Interestingly, homogentisic acid (HGA) was not observed in any of the CSF samples analysed, including those that had not received nitisinone.

In this study, the monoamine metabolites HVA, DP, VMA, tyramine, and kynurenine were not matched using the targeted feature extraction in Profinder or through manual extraction of chromatograms in MassHunter Qualitative software (Build 07.00) using the *m*/*z* of each metabolite in positive and negative polarity, respectively.

## 3. Discussion

Hypertyrosinaemia is a universal biochemical consequence of nitisinone treatment in AKU [1,2,3,4,5,6,7,8] and HT1 [9,10,11,12]. As indicated earlier, the rationale for analysing CSF from a murine model of AKU is that it gives a unique insight into the CNS that cannot be gained from the analysis of serum and urine, which provide invaluable yet indirect information.

The study presented herein is the first of its kind to report on the assessment of the CSF metabolome in a murine model of AKU. The reason for the paucity of the literature likely reflects the rarity of AKU, but also the small volume of CSF that can be collected from a mouse (~3 µL). The latter point presents a significant analytical challenge in terms of sample handling (e.g., auto samplers attached to chromatography systems) and also in that the metabolites of interest are present at picomolar to nanomolar concentrations. Moreover, the technical expertise required to collect non-blood-stained CSF from a mouse means the procedure is not frequently performed without some sample contamination.

Three metabolites were shown to have an increased abundance (log2 FC > 2.0, *p* < 0.05) following treatment with nitisinone. These changes have previously been reported following treatment with nitisinone in serum [27] and urine [23] in patients with AKU, and in the urine of a murine model of AKU [23].

The increased L-tyrosine in CSF can be explained by the increased circulating concentrations observed following treatment with nitisinone, as a consequence of its action on the HPPD enzyme. This finding is consistent with a previous study that showed nitisinone treatment to result in significant increases in L-tyrosine in brain tissue from a murine model of AKU [25].

The increased abundance of HPLA in the CSF is a curious finding, as there are no known mechanisms for its transport into the CNS, unlike L-tyrosine, which goes via the LAT-1 transporter. If HPLA transport occurred via the LAT-1 transporter, one could speculate that, due to its structural similarity to HGA, the latter would be present in a greater abundance in the CSF, which was not the case in this study. Additionally, if HPLA in CSF was a result of contamination from the circulation, then HGA would also be present at a higher abundance. The fact that the liver and kidney are the only organs that contain all of the required enzymes for the entire L-tyrosine metabolic pathway makes the local production of HPLA directly from L-tyrosine in the CNS unlikely. It is possible, however, that it may be produced via an alternative pathway in the brain that is not defined, and requires further investigation. Bernardini et al. [28] have, however, shown that the *HGD* gene is expressed in murine and human brain tissue, and reported that, in the presence of excess HGA, cultured neuronal cells produce ochronotic pigment and amyloid. The authors postulated that this may contribute to the induction of neurological complications. In contrast, Wilson et al. [29] demonstrated that the *HGD* gene was not expressed in murine brain tissue. The implications of the presence of HPLA in the CNS are unknown and require further investigation. Elevated HPLA has previously been reported in CSF from children with phenylketonuria [30] and brain tissue from a rat model of PKU [31].

Increased N-acetyl-L-tyrosine and γ-glutamyl-L-tyrosine in CSF from mice treated with nitisinone was expected, based on the observations previously reported in urine [23] and serum [27,32] from AKU patients treated with nitisinone. The large variation in abundance in this study meant the changes observed were not deemed to be statistically significant; however, the trend suggests that excess tyrosine is metabolised to N-acetyl-L-tyrosine and γ-glutamyl-L-tyrosine following treatment with nitisinone. It is presumed that, due to their structural similarity to tyrosine, both N-acetyl-L-tyrosine and γ-glutamyl-L-tyrosine are transported into the CNS via the LAT-1 transporter. Interestingly, in CSF the log2 FC of these metabolites was very similar; in serum, by contrast, the increase in N-acetyl-L-tyrosine was much greater than that of γ-glutamyl-L-tyrosine [27].

The increase in HPA is novel in CSF in the context of AKU, as its origin has been reported to be that of gut microbiota [33], as previously postulated in serum [27]. Its presence in CSF cannot be explained by this, and is likely to reflect an increased turnover of tyramine in the brain through oxidative deamination via the action of monoamine oxidase (Figure 4). Previous studies have reported low HPA levels in plasma [34], urine [35], and CSF [36] of depressed patients, suggesting a lower turnover of tyramine compared to control subjects.

In a previous study, Davison et al. [25] reported a 25-fold increase in tyramine in brain tissue from the same murine model of AKU used in this study; herein, tyramine was not detected in the CSF analysed in either group of mice. This is likely to reflect that it is present at very low concentrations and that CSF in a mouse is turned over rapidly (i.e., total CSF volume of 40 μL is turned over 12–13 times/day [37]).

Dourish et al. [38] reported p-tyramine to be present at 5.4 ng/g of mouse brain tissue and that it is metabolised rapidly by its conversion to HPA. In the previous study [25], monoamine metabolites were derivitised in brain tissue to enhance their detection. Interestingly, octopamine, one potential product of tyramine metabolism, was matched in all CSF samples herein and was not different between the experiment groups. One possible reason for this may be that tyramine in the CNS is preferentially metabolised to HPA as it is not biologically active, unlike tyramine and octopamine [39,40]. To this end, one may speculate that tyramine is present in the CSF following nitisinone, but is present below the limit of detection for LC-QTOF-MS. Previously, tyramine has been regarded as a metabolic dead-end; however, evidence suggests that it, along with other trace amines, can act as neuromodulators for monoamine neurotransmitters, altering the release of and uptake of serotonin, DP and noradrenaline [39,40]. Several studies have therefore made associations between trace amines and depression, schizophrenia, PKU, Parkinson’s disease, Reye’s syndrome, Tourette’s syndrome, epilepsy, attention deficit hyperactivity disorder, and migraines [41,42,43,44].

Reassuringly, no changes in the abundance of the other large neutral aromatic amino acids (LNAA’s) tryptophan and phenylalanine were observed in this study, suggesting that the excess L-tyrosine observed following nitisinone treatment does not compete for uptake into the CNS via the LAT-1 transporter. This contrasts with what has been proposed in PKU or HT1, where the excess of phenylalanine and L-tyrosine, respectively, have been proposed to alter neurotransmitter metabolism [17,45,46,47].

Encouragingly, MHPG (noradrenaline metabolite) and adrenaline, 5-HIAA (serotonin metabolite) and DOPAC (DP metabolite) were also not significantly altered in nitisinone-treated mice. These metabolites did, however, show large variation in their respective treatment groups, so it is important to not over-interpret this trend. In contrast, in a recent study by Pilotto et al. [47], patients with PKU were shown to have lower 5-HIAA concentrations in CSF compared to controls, suggesting that the elevated phenylalanine may contribute to altered neurotransmitter metabolism. Moreover, significant negative correlations were observed between CSF 5-HIAA, HVA, and 5-hydroxytyrptophan and phenylalanine. In a small cohort of patients with HT1, lower 5-HIAA in CSF has also been reported [17].

It is uncertain whether the high variation in metabolite abundance was a reflection of ion suppression due to the CSF matrix, or indeed a limitation of the instrumentation utilised herein. Previous studies have utilised quantitative high pressure liquid chromatography with fluorescence or electrochemical detection [48,49] for the measurement of neurotransmitters. Furthermore, monoamine metabolites have been derivitised using benzylamine to improve analytical sensitivity [50]. Targeted quantitative methods utilising liquid chromatography tandem mass spectrometry (LC-MS/MS) have also been reported, which have superior analytical specificity and sensitivity [51].

It should be borne in mind that a major limiting factor in this unique study was the very limited sample volume available for analysis. This precluded sample clean up or concentration, or indeed the application of a quantitative LC-MS/MS methodology with the use of an internal standard and matrix matched calibrators. One approach adopted by previous authors [52] has been to use sheathless capillary electrophoresis coupled to mass spectrometry, which requires nanolitre injection volumes.

Disappointingly, DP, and its major metabolites 3-MT and HVA, and the major metabolite of adrenaline, VMA, were not detected in the samples analysed. The reasons for this are not known, as these metabolites were present in the AMRT database used in this study and are likely to be present at similar concentrations to 5-HIAA, as they are in humans and have a short half-life [53]. The previous discussions regarding ion suppression from the CSF matrix and the analytical capability of the LC-QTOF-MS are equally relevant for these metabolites. Additionally, the analysis of a small volume of diluted sample may have impacted their measurement. As samples were frozen at −80 °C immediately after collection, it is unlikely that this is a reflection of metabolite degradation.

To investigate the impact of diluting CSF samples on the detection of metabolites, additional experiments were performed (*data not shown*), in which pooled CSF samples from respective treatment groups were analysed. This meant a larger sample volume was available for analysis and samples were not diluted. Disappointingly, this strategy did not improve the detection of 3-MT, HVA, tyramine, or VMA, meaning that the impact of hypertyrosinaemia on DP metabolism in CSF could not be fully evaluated.

Further work is required to assess the impact of hypertyrosinaemia on both DP and serotonin-related metabolites in CSF, using a targeted quantitative method such as LC-MS/MS. This will, however, not be without its own challenges, due to the limited volume of CSF that is available from a mouse. An alternative animal model could also be considered, for example the rat, where larger volumes of CSF can be obtained for analysis as the total CSF volume is estimated to be 250 µL [54].

## 4. Materials and Methods

### 4.1. Reagents

Water for mobile phases was purified in-house (DIRECT-Q 3UV Millipore water purification system). Methanol, acetonitrile, and isopropanol were purchased from Sigma Aldrich (Dorset, UK). Formic acid and ammonium formate were obtained from Biosolve, (Valkenswaard, The Netherlands). All reagents were LC-MS grade. The 2 mL amber screw vial and 150 µL glass inserts with polymer feet (maximum recovery vials) were purchased from Agilent (Cheadle, UK) and artificial CSF from Harvard apparatus (Holliston, MA, USA). All analytical standards were obtained from Sigma (UK) as 1 mg/mL Cerilliant standards.

### 4.2. Animal Experiments and CSF Sample Collection

A murine model of AKU was used for all experiments, as described previously (Preston et al., 2014). A total of seventeen BALB/c *Hgd*^−/−^ mice (Figure 5 for age and gender) were included in this study; eight mice were administered nitisinone (4 mg/L) through their drinking water for 6 days, the remaining nine mice received no nitisinone. All animals were housed in air-conditioned rooms (with a 12 h dark/light cycle) at 20 °C and 53% humidity, with access to food and water ad libitum, at Liverpool John Moores University. All animal experiments complied with the ARRIVE guidelines and were carried out in accordance with the U.K. Animals (Scientific Procedures) Act, 1986 and associated guidelines, EU Directive 2010/63/EU for animal experiments.

All mice were culled by carbon dioxide asphyxiation and CSF was removed from the cisterna magna following the puncture technique of Liu et al. [55], with some modifications. The cisterna magna was exposed by dissecting the skin and overlaying muscles. Cauterisation was used to dry up any bleeding from surrounding tissues. A pulled glass pipette with an internal diameter of ~0.4 mm was attached to silicone tubing and connected to a 1 mL syringe with a 19 g needle. The tip of the glass pipette was used to puncture the membrane and held still just below the membrane by one person. Through a double-headed microscope, a second person then used the 1 mL syringe to apply gentle pressure to encourage the CSF to flow up the capillary tube. Only clear CSF samples (i.e., non-blood-stained) were collected. Once collected, the CSF samples were transferred to clear Eppendorf tubes and stored at −80 °C until analysis. Samples were not acidified. The same glass pipette was used to collect CSF from each animal, and was washed with pure water between each animal.

### 4.3. Murine and Quality Control Sample Preparation

Murine CSF samples were prepared by diluting 2 µL CSF with 2 µL deionized water (DIRECT-Q 3UV Millipore water purification system). CSF samples were pipetted directly into a 150 µL glass insert with polymer feet, which sat in a 2 mL amber screw vial (Agilent, Cheadle, UK). These vials were used to ensure maximal recovery of samples. Deionised water was also added directly to the glass insert, and mixed with the CSF sample using a pipette. The glass vials were centrifuged at 600 rpm for 10 min to ensure the diluted sample was at the bottom of the vial for sampling. Then, 1 µL of diluted CSF was analysed in negative and positive polarities, respectively.

As the CSF sample volume was very limited (~3 µL per mouse), biological group QCs were not prepared. The performance of the analytical run was assessed using system quality control samples that were prepared in artificial CSF (Harvard Apparatus, Holliston, MA, USA). These contained phenylalanine (100 µmol/L), L-tyrosine (100 µmol/L), nitisinone (10 µmol/L), and succinylacetone (30 µmol/L). Phenylalanine, L-tyrosine, and nitisinone were assessed in positive polarity and succinylacetone in negative polarity. Each analytical run (i.e., negative and positive polarities) commenced with 20 replicate injections of the system quality control sample to condition the system.

The order of individual samples was randomised computationally. System QC samples were interspersed throughout the analytical sequence every sixth injection, to assess the analytical system performance. Binary pump pressures and reference ions were also monitored to assess the performance of the LC-QTOF-MS (Appendix A).

### 4.4. Analytical Method

#### 4.4.1. LC-QTOF-MS Conditions

LC-QTOF-MS conditions used for the analysis of murine CSF samples were the same as those previously reported for the analysis of urine [23] and serum samples [27] (see Appendix A).

#### 4.4.2. Data Acquisition and Handling Parameters

Data were acquired using Acquisition (Build 06.00, Agilent, Cheadle, UK). Quality checks and processing of raw data files (Agilent ‘.d’ files) were performed with Qualitative Analysis software (Build 07.00, Agilent, Cheadle, UK).

Extracted ion chromatograms of reference masses were performed to check mass accuracy remained <5 ppm throughout the run, and that the reference ion signal did not drop out during the chromatographic run. In addition, to check chromatographic reproducibility, binary pump pressure curves for injections across each analytical sequence were overlaid. Mass accuracy and chromatographic reproducibility were acceptable for all experiments performed.

The acquired profiling sample data were mined for signals against an established in-house AMRT database of compounds (which contains theoretical accurate mass, measured retention time, and empirical formula for 469 intermediary metabolites, MW 72–785) [23,26,27] using ‘targeted feature extraction’ with Profinder software (Build 08.00, Agilent, Cheadle, UK). Targeted feature extraction uses the molecular formulae from the AMRT database to extract and group spectral signals (i.e., adducts, isotopes and multimers) that correspond to individual database compounds. Feature extraction employed a window of theoretical accurate mass ±10 ppm and database retention time ±0.3 min. Allowed species were: H^+^, Na^+^ and NH4+ (positive polarity) and H- and CHO^2−^ (negative polarity). Dimers were allowed for both polarities. The charge state range was 1–2. Data files were then exported from Profinder as ‘.CEF’ files as a whole batch for each profiling experiment, and imported to Mass Profiler Professional (MPP) software for statistical analysis (Build 14.5, Agilent, Cheadle, UK).

#### 4.4.3. Data Quality Control and Statistical Analysis

In the absence of biological group QC samples (due to low CSF sample volume), analytical performance was assessed using spiked artificial CSF samples. Criteria for deeming the analytical run as acceptable were: CV of raw area abundance for target compounds <15%; mean error from target mass <5 ppm; and mean difference from target retention time <0.3 min.

In MPP, sample data files were scaled to the median of all samples, and chemical entities were filtered based on their frequency (i.e., present in 100% of samples in either group) and variability across the experimental groups (CV < 25%). Statistical analyses performed in MPP were based on compound signal intensity, expressed as total peak area. Profiles from the experimental group that received nitisinone were compared to the profiles that did not receive nitisinone using a moderated t-test with Benjamini–Hochberg false-discovery rate adjustment. Log2 fold change was calculated between each experimental group, and this was based on raw peak area data. Changes were deemed significant if the asymptotic *p* value was <0.05 and a log2 FC >2.0. Principal component analysis (PCA) employing a 4-component model was performed on each filtered dataset using MPP.

## 5. Conclusions

This is the first study to report the impact of treatment with nitisinone on the CSF metabolome, and specifically on monoamine neurotransmitter metabolites. This study showed that L-tyrosine and a small number of related metabolites had a greater abundance in mice that were treated with nitisinone. Of particular interest is the observation that the abundance of HPA increased following treatment with nitisinone, suggesting that there may be an increase in the turnover of the trace amine tyramine. Further work is required to understand the significance of these findings and the mechanisms by which the altered metabolite abundances occur. This in turn will help inform the implications these changes may have in patients with hypertyrosinaemia.

## Figures and Tables

**Figure 1 metabolites-12-00477-f001:**
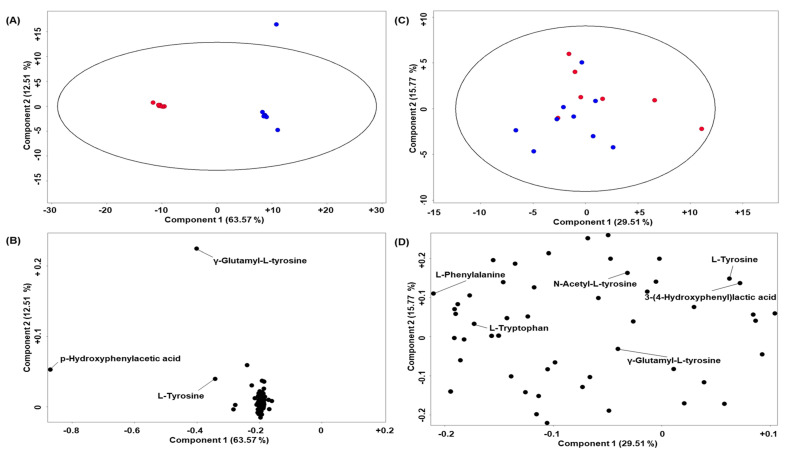
Principal component analysis (PCA) to assess the effect of nitisinone therapy on metabolite profiles in CSF samples collected from AKU mice. Blue circle—no treatment and red circle—nitisinone therapy (4 mg/L, six days). PCA plots (**A**) in positive and (**C**) negative polarity. Ellipse around data points—95 % confidence interval. Loadings plots (**B**,**D**) show the respective contributions of individual metabolites to components 1 and 2 in positive and negative polarity, respectively.

**Figure 2 metabolites-12-00477-f002:**
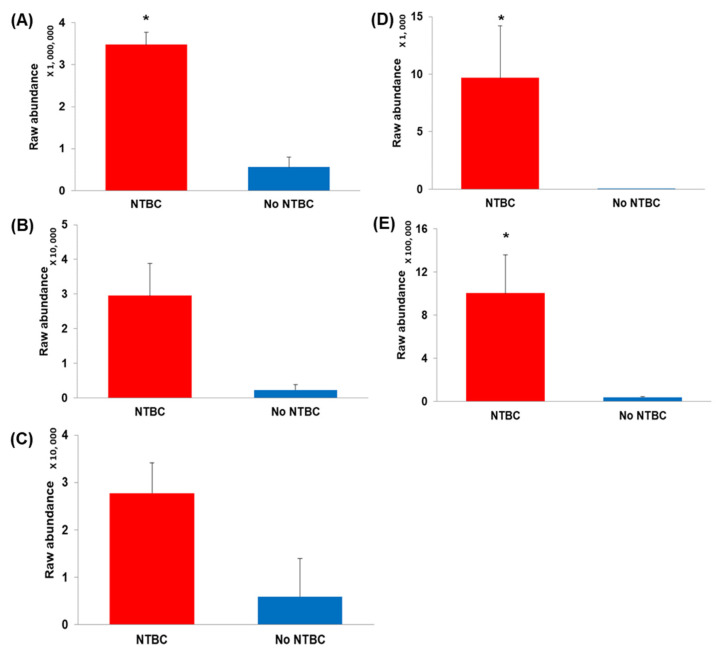
Bar graphs showing the mean raw abundances from LC-QTOF-MS analysis for metabolites that were matched to the AMRT database and had a log2 FC > 2.0. Error bars represent +1 standard deviation of the mean raw metabolite abundance (**A**) L-Tyrosine; (**B**) N-Acetyl-L-tyrosine; (**C**) γ-Glutamyl-L-tyrosine; (**D**) HPA; (**E**) HPLA. *—indicates significance, *p* < 0.05. NTBC—nitisinone. All data presented based on positive polarity (*n* = 9 No NTBC, *n* = 8 NTBC) except (**E**) which is from negative polarity (*n* = 9 No NTBC, *n* = 7 NTBC).

**Figure 3 metabolites-12-00477-f003:**
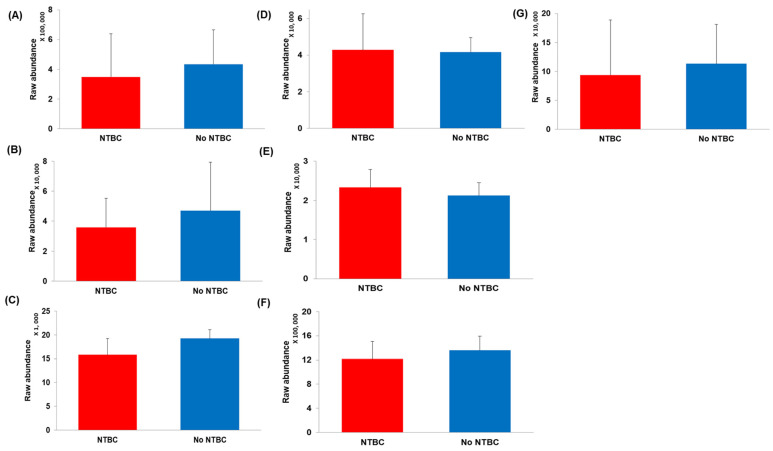
Bar graphs showing mean raw abundances from LC-QTOF-MS analysis for AMRT-matched metabolites that are important in neurotransmitter metabolism. Error bars represent +1 standard deviation of the mean raw metabolite abundance. The abundance of these metabolites between the two experimental groups had a log2 FC < 2.0 and there were no significant differences. (**A**) Tryptophan, (**B**) 5-HIAA, (**C**) Octopamine, (**D**) Adrenaline, (**E**) MHPG, (**F**) Phenylalanine, (**G**) DOPAC. It is important to note that 5-HIAA and DOPAC showed large variability within each experimental group (i.e., CV > 25%) and were excluded from further statistical analysis. NTBC—nitisinone. All data presented based on positive polarity (*n* = 9, No NTBC and *n* = 8 NTBC).

**Figure 4 metabolites-12-00477-f004:**
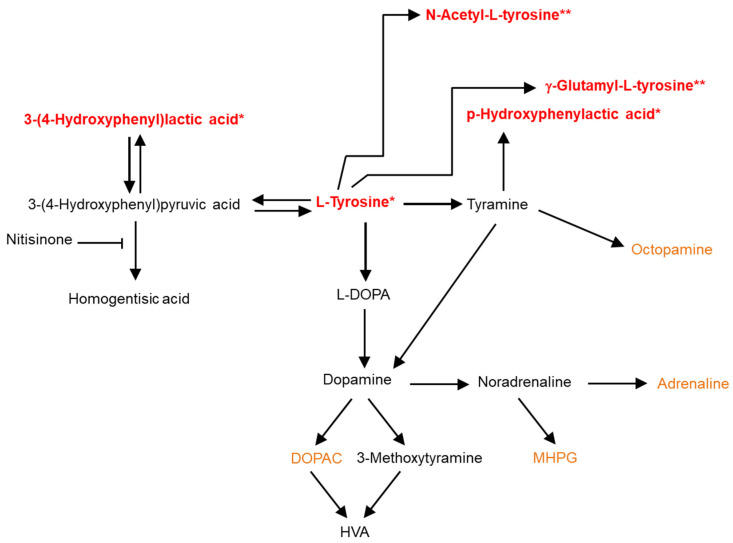
Summary of metabolite differences that were observed between *Hgd*^−/−^ mice that received no treatment and *Hgd*^−/−^ mice that received treatment with nitisinone. Red—compounds that had a higher abundance in CSF from mice treated with nitisinone; orange—metabolites matched in CSF samples, but abundances were not significantly different between the two groups of mice; black—compounds not matched in samples. *—log2 FC > 2.0, *p* < 0.05; **—log2 FC > 2.0, *p* > 0.05. HVA—homovanillic acid; DOPAC—dihydroxyphenylacetic acid; L-DOPA—L-3,4-dihydroxyphenylalanine; MHPG—3-methoxy-4-hydroxyphenylglycol.

**Figure 5 metabolites-12-00477-f005:**
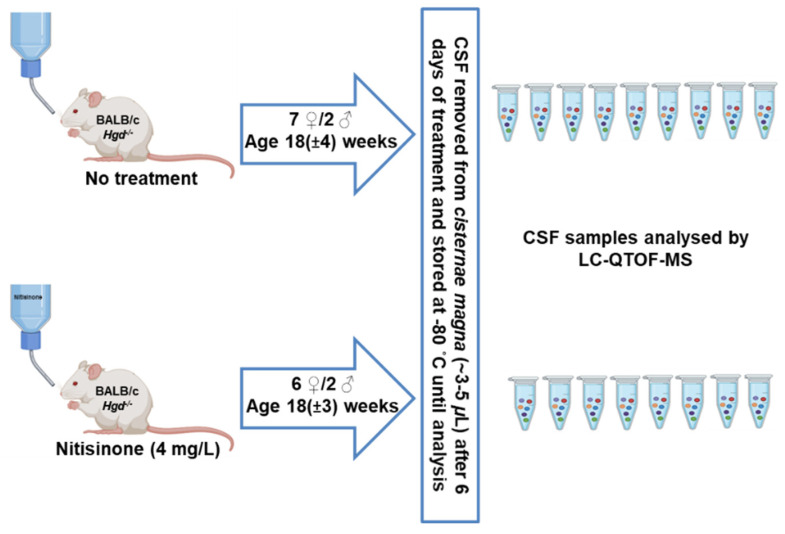
Summary of experimental workflow involving murine model of AKU. This includes mouse gender (♂—male; ♀—female) and mean age (±standard deviation), treatment groupings, and sample collection and storage. Mass spectrometry analysis was performed in positive and negative polarities in all samples collected, except one sample in the negative polarity, due to a low sample volume (i.e., *n* = 7/8 samples were analysed in the nitisinone group).

**Table 1 metabolites-12-00477-t001:** Summary of results from the analysis of quality control samples (*n* = 8) throughout analytical runs. RT—retention time; CV—coefficient of variation; ppm—parts per million.

Compound	CV of Raw Area Abundance (%)	Mean Error from Target Mass (ppm)	Mean Difference from Target RT (min)
**L-Tyrosine (100 µmol/L)**	8.8	0.69	0.07
**L-Phenylalanine (100 µmol/L)**	2.4	0.20	0.06
**Nitisinone (10 µmol/L)**	11.9	0.26	0.004
**Succinylacetone (30 µmol/L)**	4.6	0.42	0.02

**Table 2 metabolites-12-00477-t002:** CSF metabolite changes identified post-nitisinone therapy in BALB/c *Hgd*^−/−^ mice using an in-house AMRT database. Abundance expressed as log2 FC, compared to BALB/c *Hgd*^−/−^ mice that did not receive nitisinone. Log2 FC included if abundance >2.0 and *p* < 0.05. *—increase in abundance observed in positive and negative polarities, data included represent the lowest log2 FC, which was observed in negative polarity; ^†^—increase in abundance observed in positive polarity; ^††^—increase in abundance observed in negative polarity.

Compound	Log2 FC	Abundance		*p* Value	Metabolic Pathway Affected
		Up	Down		
p-Hydroxyphenylacetic acid (HPA) ^†^	4.0	√		6.37 × 10^−19^	Tyramine
3-(4-Hydroxyphenyl)lactic acid (HPLA) ^††^	3.9	√		6.15 × 10^−11^	L-Tyrosine
L-Tyrosine *	2.6	√		8.7 × 10^−10^	L-Tyrosine

## Data Availability

Data supporting reported results can be found at Metabolomics Workbench: https://www.metabolomicsworkbench.org/. Study ID: adavison_20220505_020022_mwtab.txt. Accessed 5 May 2022.

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
