# Peer review of "Impact of Nitisinone on the Cerebrospinal Fluid Metabolome of a Murine Model of Alkaptonuria"

_metabolites, 2022, doi:10.3390/metabo12060477_

Round 1

Reviewer 1 Report

In this study Davison et al., report the impact of nitisinone administration on CSF amino acids, related to dopamine, in a murine model of alkaptonuria. LC-QTOF-MS results report increased CSF tyrosine and tyrosine metabolites in treatment group. Although the results of this study were unsurprising, this group importantly record AA alterations in the CSF of nitisinone treament in AKU mice. Study design, statistical approach and analyses are sound.

Minor consideration: Graphical depiction in figure one is suboptimal with limited legibility.

Author Response

Dear Reviewer 1,

Thank you for your comments. I have taken them on board and revised the manuscript accordingly. Specifically I have responded to:

Reviewer comment: Graphical depiction in figure 1 is suboptimal with limited legibility.

Response: Figure 1 revised. Also revised Figures 4 and 5 to improve consistency of terminology (Figure 4) and to make design in keeping with graphical abstract (Figure 5).

Best wishes,

Andrew Davison

Reviewer 2 Report

Dear Editor,

The paper by Davison entitled: “Impact of nitisinone on the cerebrospinal fluid metabolome of a 2 murine model of alkaptonuria” is a n original paper focused to evaluate the impact of nitisinone on the cerebrospinal fluid (CSF) metabolome in a murine model of AKU, with a view to  providing additional insight into metabolic changes that occur following treatment with nitisinone.

The article is considerable scientific interest; in my opinion, it will be appreciated by readers.

The abstract summarizes and reflect the work described

The tables and figures reflect the work described in the paper.

The references are appropriate.

Major revision

 -the method section may follow  introduction

-An extensive English editing is needed. 

- after discussion may be reported the conclusion that may be extend with some hypothesis in the future in clinical bases

- please create a graphic abstract

Author Response

Dear Reviewer 2,

Thank you for your comments. I have taken them on board and revised the manuscript accordingly. Specifically I have responded to:

Reviewer comment: the method section may follow introduction

Response: The layout of the paper follows the journal style (i.e. introduction then results).

Reviewer comment: An extensive English editing is needed.

Response: manuscript reviewed and minor typographical and stylistic changes made. Not always tracked as involved deleting text, changes highlighted in yellow.

Reviewer comment:  after discussion may be reported the conclusion that may be extend with some hypothesis in the future in clinical bases

Response: manuscript reviewed and edited, changes highlighted in yellow.

Reviewer comment:  please create a graphic abstract

Response: Graphical abstract made (end of manuscript).

In addition resolution and quality of Figures 1, 4 and 5 improved.

Best wishes,

Andrew Davison
